# Nanophotonic Pockels modulators on a silicon nitride platform

Koen Alexander [1,2], John P. George[1,2,3], Jochem Verbist[1,2,4], Kristiaan Neyts [2,3], Bart Kuyken[1,2], Dries Van Thourhout [1,2] & Jeroen Beeckman [2,3]

Silicon nitride (SiN) is emerging as a competitive platform for CMOS-compatible integrated photonics. However, active devices such as modulators are scarce and still lack in performance. Ideally, such a modulator should have a high bandwidth, good modulation efficiency, low loss, and cover a wide wavelength range. Here, we demonstrate the first electro-optic modulators based on ferroelectric lead zirconate titanate (PZT) films on SiN, in both the O-band and C-band. Bias-free operation, bandwidths beyond 33 GHz and data rates of 40 Gbps are shown, as well as low propagation losses ($\alpha \approx 1\,\mathrm{dB\,cm^{-1}}$). A half-wave voltage-length product of 3.2 V cm is measured. Simulations indicate that further improvement is possible. This approach offers a much-anticipated route towards high-performance phase modulators on SiN.

[1] Photonics Research Group, INTEC Department, Ghent University-imec, Technologiepark-Zwijnaarde 15, 9052 Zwijnaarde, Belgium. [2] Center for Nano- and Biophotonics (NB-Photonics), Ghent University, Technologiepark-Zwijnaarde 15, 9052 Zwijnaarde, Belgium. [3] Liquid Crystals and Photonics Group, ELIS Department, Ghent University, Technologiepark-Zwijnaarde 15, 9052 Zwijnaarde, Belgium. [4] IDLab, INTEC Department, Ghent University-imec, Technologiepark-Zwijnaarde 15, 9052 Zwijnaarde, Belgium. These authors contributed equally: Koen Alexander, John P. George. Correspondence and requests for materials should be addressed to D.V.T (email: Dries.VanThourhout@UGent.be) or to J.B. (email: Jeroen.Beeckman@UGent.be)

The exponential increase in data traffic requires high-capacity optical links. A fast, compact, energy-efficient, broadband optical modulator is a vital part of such a system. Modulators integrated with silicon (Si) or silicon nitride (SiN) platforms are especially promising, as they leverage complementary-metal-oxide-semiconductor (CMOS) fabrication techniques. This enables high-yield, low-cost, and scalable photonics, and a route towards co-integration with electronics[1]. SiN-based integrated platforms offer some added advantages compared to silicon-on-insulator, such as a broader transparency range[2], a lower propagation loss[3,4], significantly lower nonlinear losses[2,5], and a much smaller thermo-optic coefficient[2]. Therefore, phase modulators on SiN in particular would open new doors in other fields as well, such as nonlinear and quantum optics[5–7], microwave photonics[8], optical phased arrays for LIDAR or free-space communications[9], and more.

State-of-the-art silicon modulators rely on phase modulation through free carrier plasma dispersion in p–n[10], p–i–n[11], and MOS[12] junctions. Despite being relatively fast and efficient, these devices suffer from spurious amplitude modulation and high insertion losses. Alternative approaches are based on heterogeneous integration with materials such as III–V semiconductors[13,14], graphene[15,16], electro-optic organic layers[17], germanium[18], or epitaxial BaTiO$_3$ (BTO)[19–21].

Most of these solutions are not viable using SiN. Due to its insulating nature, plasma dispersion effects and many approaches based on co-integration with III–V semiconductors, graphene, and organics, which rely on the conductivity of doped silicon waveguides, cannot be used. The inherent nature of deposited SiN further excludes solutions using epitaxial integration. Finally, SiN is centrosymmetric, hampering Pockels-based modulation in the waveguide core itself, in contrast to aluminum nitride[22], or lithium niobate[23]. Nonetheless, modulators on SiN exist. Using double-layer graphene, Phare et al.[24] achieved high-speed electro-absorption modulation, and using piezoelectric lead zirconate titanate (PZT) thin films, phase modulators based on stress-optic effects[25], and geometric deformation[26], have been demonstrated, albeit with sub-MHz electrical bandwidth.

In this work, we use a novel approach for co-integration of thin-film PZT on SiN[27]. An intermediate low-loss lanthanide-based layer is used as a seed for the PZT deposition, as opposed to the highly absorbing Pt-based seed layers used conventionally[25,26], enabling direct deposition of the layer on top of SiN waveguides fabricated using front-end-of-line CMOS processes.

We demonstrate efficient high-speed phase modulators on a SiN platform, with bias-free operation, modulation bandwidths exceeding 33 GHz in both the O-band and C-band, and data rates up to 40 Gbps. We measure propagation losses down to 1 dB cm$^{-1}$ and half-wave voltage-length products $V_\pi L$ down to 3.2 Vcm for the PZT-on-SiN waveguides. Moreover, based on simulations we argue that the $V_\pi L$ can be considerably reduced by optimizing the waveguide cross-section, without significantly increasing the propagation loss. Hence, the platform provides an excellent trade-off between optical losses and modulation efficiency. According to simulations, the product $V_\pi L\alpha$ can be as low as 2 VdB in optimized structures. Pure phase modulation also enables complex encoding schemes (such as QPSK), which are not easily achievable with absorption modulation. These results, especially in terms of the achieved modulation bandwidths, strongly improve upon what is currently possible in SiN[25,26]. In terms of $V_\pi L\alpha$, this platform can furthermore improve upon carrier dispersion modulators in silicon-on-insulator, which suffer from inherent carrier-induced losses absent in Pockels modulators[28].

## Results

**Device design and fabrication.** The waveguides are patterned using 193 nm deep ultraviolet lithography in a 330-nm-thick layer of low pressure chemical vapor deposited SiN on a 3.3-µm-thick buried oxide layer, in a CMOS pilot line. Subsequently, plasma-enhanced chemical vapor deposited SiO$_2$ (thickness ≈1 µm) is deposited over the devices and planarized, either using a combination of dry and wet etching, or by chemical–mechanical polishing (CMP). This step is performed so that the top surface of the SiN waveguide and the surrounding oxide are coplanar. The PZT films are deposited by chemical solution deposition (CSD), using a lanthanide-based intermediate layer (see Methods and ref.[27]). Finally, Ti/Au electrical contacts are patterned in the vicinity of the waveguides using photolithghraphy, thermal evaporation, and lift-off. For the samples planarized through CMP, propagation losses of 1 dB cm$^{-1}$ are measured on PZT-covered waveguides without metallic contacts (see Supplementary Note 2).

Figure 1a, b show the top view and waveguide cross-section of a C-band ring modulator, and for images of the other fabricated modulators (O-band ring, C-band Mach–Zehnder), see Supplementary Note 1. Figure 1c shows a schematic of the cross-section. An electric field is applied through in-plane electrodes, changing the refractive index in the PZT and hence the effective index of the waveguide mode. The PZT thin films exhibit a higher refractive index ($n \approx 2.3$) than SiN ($n \approx 2$), so a significant portion of the optical mode is confined in the PZT. A grating coupler is used for incoupling and outcoupling, into the fundamental quasi-transverse electric (quasi-TE) optical mode. The combined loss of a grating coupler and the transition between a bare and PZT-covered waveguide section is ≈12 dB at the optimum, with a 3 dB bandwidth of ≈90 nm. However, this is currently not optimized and can still be improved by design.

**DC characterization and poling stability.** Figure 1d shows the transmission spectrum of a C-band (1530–1565 nm) ring modulator. The ring has a loaded $Q$ factor of 2230 and a free spectral range $\Delta\lambda_{FSR} \approx 1.7$ nm. The ring radius, the length of the phase shifter $L$, and the electrode spacing are, respectively, 100, 524, and 4.4 µm. The relatively low $Q$ factor is caused by sub-optimal alignment of the electrodes.

After deposition, the PZT crystallites have one crystal plane parallel to the substrate, but no preferential orientation in the chip's plane. To obtain a significant electro-optic response for the quasi-TE optical mode, a poling step is performed by applying 60–80 V (≈150 kV cm$^{-1}$) for 1 h at room temperature, followed by several hours of stabilization time.

The transmission spectrum is measured for different direct current (DC) voltages applied across the PZT layer (Fig. 1e). The voltage-induced index change shifts the resonance. In Fig. 1f, the resonance wavelength shift is plotted as a function of voltage, and the slope gives the tuning efficiency $\Delta\lambda/\Delta V = -13.4$ pm V$^{-1}$. From this, we estimate the half-wave voltage-length product to be $V_\pi L = |L\lambda_{FSR}\Delta V/(2\Delta\lambda)| \approx 3.3$ Vcm. Through simulation of the optical mode and DC electric field, the effective electro-optic coefficient $r_{eff}$ of the PZT layer is estimated to be 61 pm V$^{-1}$ (see Methods). Measurements on other modulator structures yield consistent values for $r_{eff}$ (67 and 70 pm V$^{-1}$ for, respectively, the C-band Mach–Zehnder and O-band ring), and the smallest $V_\pi L$ value (≈3.2 V cm) was measured on an O-band ring (Supplementary Note 1).

The PZT was poled prior to the measurements, after which no bias voltage was used. To demonstrate longer term stability of the poling, the DC tuning efficiency was periodically measured

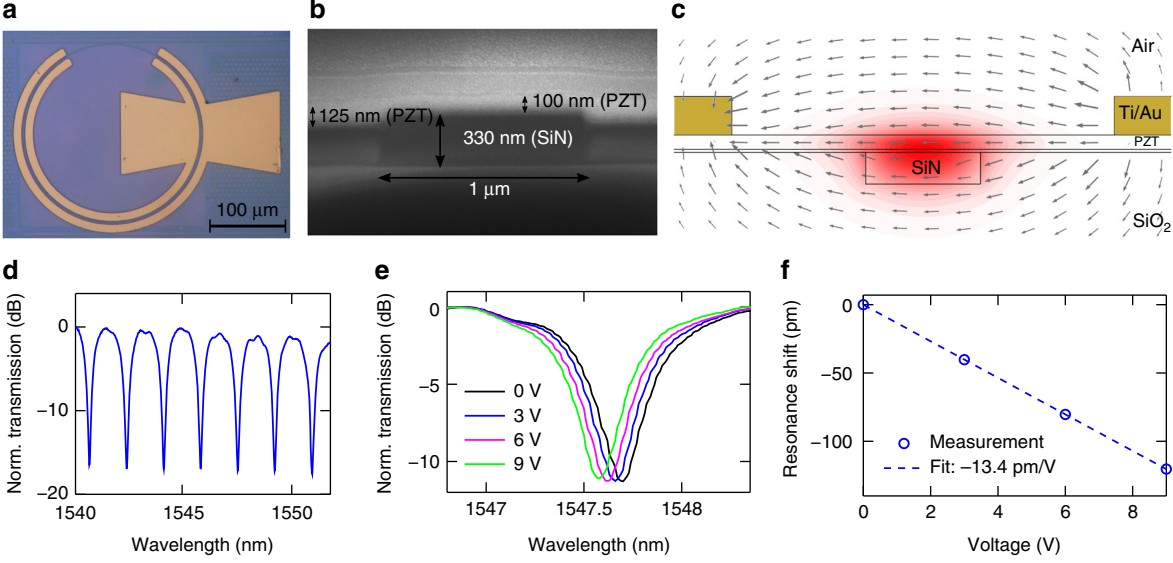

**Fig. 1** Design and static response of a C-band ring modulator. **a** Top view of a PZT-on-SiN ring modulator. **b** Cross-section of a PZT-covered SiN waveguide. The image contrast was enhanced for clarity. **c** Schematic of the PZT-covered SiN waveguide. The fundamental TE optical mode is plotted in red. The quiver plot shows the applied electric field distribution between the electrodes. PZT thickness, waveguide width, and gap between the electrodes are, respectively, 150 nm, 1200 nm, and 4 μm. **d** Normalized transmission spectrum of a C-band ring modulator. **e** Transmission spectra for different DC voltages. **f** Resonance wavelength shift versus voltage applied across the PZT, including a linear fit

(sweeping the voltage over $[-2, +2]$ V) on a C-band ring over a total time of almost 3 days. In Fig. 2, the absolute value of the resulting tuning efficiency $\Delta\lambda/\Delta V$ is plotted as a function of time, decaying towards a stable value of about 13.5 pm V$^{-1}$ over the course of several hours. The poling stabilized and there have been no indications of decay over much longer periods of time, hence modulation is possible without a constant bias, as opposed to similar materials like BTO[19–21].

**High-speed characterization.** For many applications, high-speed operation is essential. In Fig. 3a the setup used for high-speed characterization is shown. In Fig. 3b, the $|S_{21}|$ measurement for different modulators is plotted. The measured 3 dB bandwidths of both rings are around 33 GHz, and the Mach–Zehnder has a bandwidth of 27 GHz. The bandwidths are not limited by the intrinsic material response of PZT, but by device design and/or characterization equipment, as the dominating contributions to the Pockels effect are expected to have a bandwidth which is almost two orders of magnitude larger[29,30]. We furthermore demonstrate that our platform can be used for high-speed data transmission. In Fig. 3c, eye diagrams are plotted for different bitrates, a non-return-to-zero (NRZ) binary sequence (4.2 V peak-to-peak) is used. The eye remains open up until about 40 Gbps, limited by the arbitrary waveform generator (AWG) (25 GHz bandwidth), rather than by the modulator itself. The bit error rates were estimated from the measured eye diagrams[31], and are below the hard-decision forward error coding limit of $3.8 \times 10^{-3}$ [32,33] for bitrates up to 40 Gbps (see Supplementary Note 3). At 10 Gbps, an extinction ratio of 3.1 dB is measured (see Supplementary Note 4).

**Device optimization.** The presented devices were not fully optimized in terms of electro-optic modulation parameters. Primarily the PZT thickness could be increased. Sub-optimal thicknesses were used to reduce bend losses and coupling losses into PZT-covered waveguide sections. These limitations can be alleviated by device design. In Fig. 4, simulation results of the

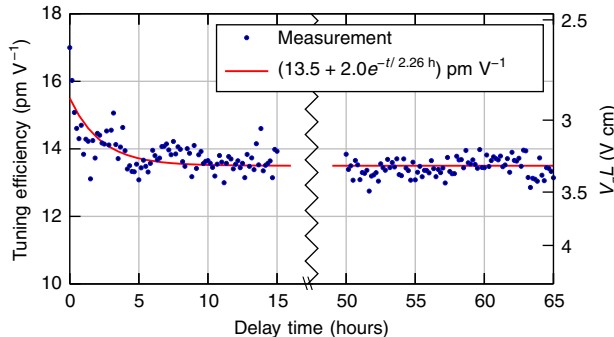

**Fig. 2** Poling stability of the electro-optic film. Tuning efficiency (C-band ring) as a function of time after poling. The axis on the right shows the estimated corresponding $V_\pi L$

most important figures of merit are plotted as a function of the PZT layer thickness and of the electrode spacing. Waveguide height, width, and the wavelength are, respectively, 300 nm, 1.2 μm, and 1550 nm. The waveguide propagation loss $\alpha$ (Fig. 4a) is calculated as the sum of a contribution caused by the electrodes, and a constant intrinsic propagation loss of 1 dB cm$^{-1}$, a realistic value if the samples are planarized using CMP (see Supplementary Note 2). The half-wave voltage-length product $V_\pi L$ (see Methods) and the product $V_\pi L\alpha$ are shown in Fig. 4b, c, respectively. $V_\pi L$ represents a trade-off between drive voltage and device length, $V_\pi L\alpha$ also takes into account loss, and is arguably more important for many applications[26]. The loss increases with decreasing electrode spacing, but also with increasing PZT thickness, since the mode expands laterally. Due to the increasing overlap between the optical mode and the PZT, $V_\pi L$ decreases with increasing thickness. $V_\pi L$ also increases with increasing electrode spacing. An optimization of the waveguide width is given in the Supplementary Note 5. From Fig. 4b it is clear that $V_\pi L$ can go well below 2 Vcm. The interplay between these

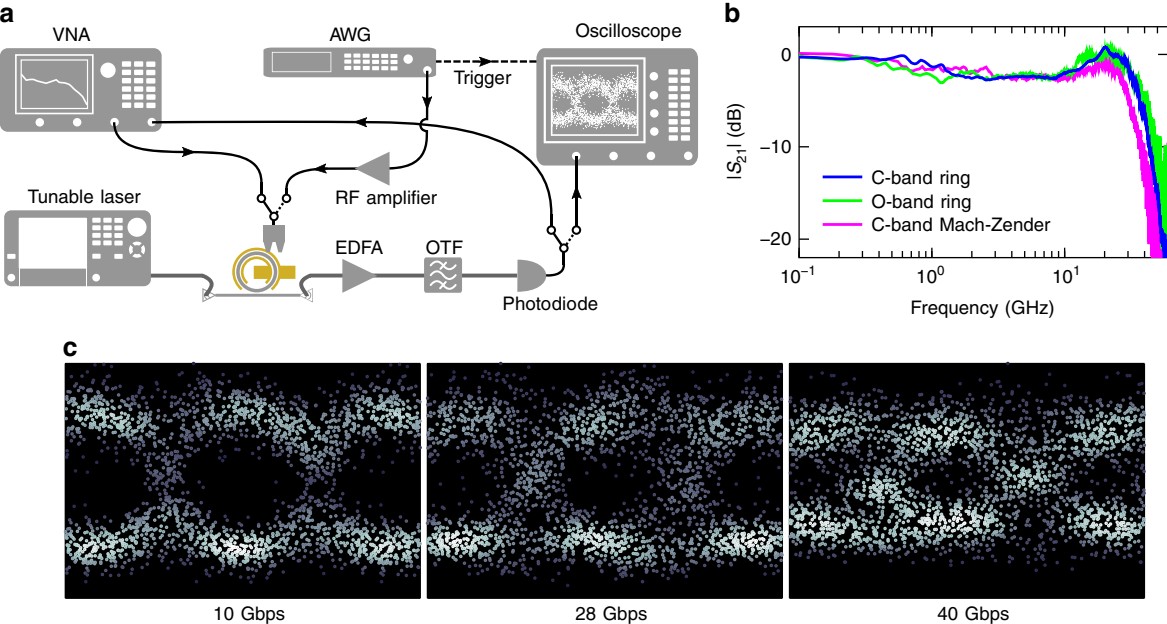

**Fig. 3** High-speed measurements. **a** Sketch of the setup used for small signal measurements (solid path in the switches) and for the eye diagram measurements (dashed path). VNA: vector network analyzer, AWG: arbitrary waveform generator, OTF: optical tunable filter. **b** Electro-optic small signal ($|S_{21}|$ parameter) measurement of several modulators. **c** Eye diagrams of a C-band ring modulator, measured with a non-return-to-zero scheme ($2^9 - 1$ pseudorandom binary sequence) and a peak-to-peak drive voltage of 4.2 V

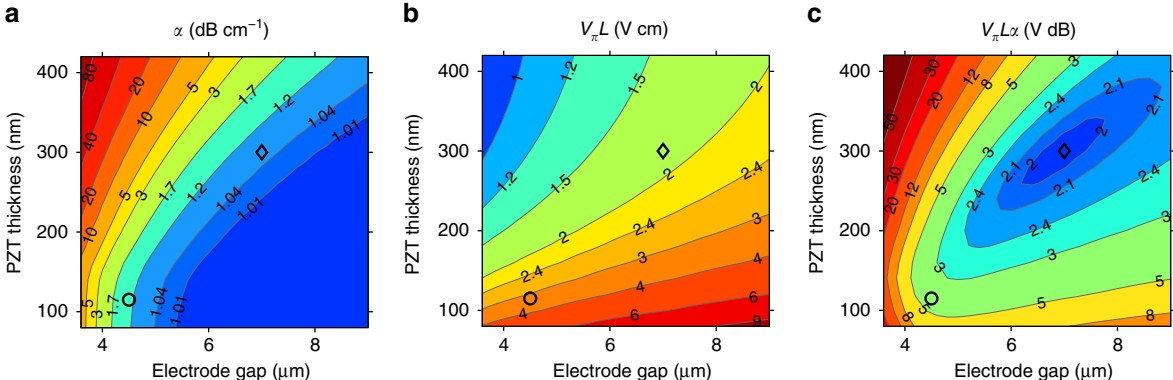

**Fig. 4** Numerical optimization of a PZT-on-SiN phase modulator. Simulation of the waveguide loss $\alpha$ (**a**), the half-wave voltage-length product $V_\pi L$ (**b**), and their product $V_\pi L\alpha$ (**c**) of a PZT-covered SiN waveguide modulator of the type shown in Fig. 1c, for a wavelength of 1550 nm. Waveguide height, width, and intermediate layer thickness are, respectively, 300 nm, 1.2 μm, and 20 nm. The intrinsic waveguide loss (in the absence of electrodes) was taken to be 1 dB cm$^{-1}$, and the effective electro-optic Pockels coefficient was 67 pm V$^{-1}$. The circles show the approximate parameters used in this work, and the diamonds show the optimal point with respect to $V_\pi L\alpha$

different dependencies can be seen in the plot of $V_\pi L\alpha$ (Fig. 4b), which has an optimum for which $V_\pi L\alpha \approx 2$ VdB.

## Discussion

To conclude, we have demonstrated a novel platform for efficient, optically broadband, high-speed, nanophotonic electro-optic modulators. Using a relatively simple chemical solution deposition procedure, we incorporated a thin film of strongly electro-optic PZT onto a SiN-based photonic chip. We demonstrated stable poling of the electro-optic material, and efficient and high-speed modulation, in the absence of a bias voltage. O-band and C-band operation was shown; however, we expect the platform to be operational into the visible wavelength range (>450 nm)[2,34,35]. From simulations it is clear that the devices characterized in this

paper do not yet represent the limitations of the platform and $V_\pi L\alpha \approx 2$ V dB is achievable. Moreover, our approach is unique in its versatility, as the PZT film can be deposited on any sufficiently flat surface, enabling the incorporation of the electro-optic films onto other guided-wave platforms.

## Methods

**PZT deposition and patterning.** While the details of the lanthanide-assisted deposition procedure have been published elsewhere[27], a short summary is given here. Intermediate seed layers based on lanthanides are deposited prior to the PZT deposition. The intermediate layer acts as a barrier layer to prevent the inter-diffusion of elements and as a seed layer providing the lattice match to grow highly oriented thin films. A critical thickness of the intermediate layer needs to be maintained (>5 nm) to avoid diffusion and secondary phase formations. However, on samples with considerable surface topology, thicker intermediate layers are necessary to provide good step coverage and to avoid any issues associated with the

conformity in spin-coating. On our samples planarized through etching, step heights between oxide and SiN waveguides varied. We typically used an intermediate layer of thickness ≈24 nm to avoid issues. Both the intermediate layer and the PZT thin films are deposited by repeating the spin-coating and annealing procedure, which allows easy control of the film thickness. The PZT layer is deposited and annealed at 620 °C for 15 min in a tube furnace under an oxygen ambient. This CSD method, also called sol–gel, provides a cheap and flexible alternative to achieve high-quality stoichiometric PZT thin films regardless of the substrate material. A reactive ion etching procedure based on $SF_6$ chemistry is used to pattern the PZT layer. The PZT film was removed selectively over the grating couplers used for the optical measurements.

**High-speed measurements.** The small-signal response measurements were performed using an Agilent PNA-X N5247A network analyzer and a high-speed photodiode (Discovery Semiconductors DSC10H Optical Receiver). For the eye diagram measurements, an arbitrary waveform generator (Keysight AWG M8195A) and radio frequency (RF) amplifier (SHF S807) are used to apply a pseudorandom NRZ binary sequence, the modulator output is measured with a Keysight 86100D oscilloscope with 50 GHz bandwidth and Discovery Semiconductors DSC-R409 PIN-TIA Optical Receiver.

**Calculation of the electro-optic parameters.** Using COMSOL Multiphysics®, several parameters can be calculated that strongly influence the performance of the modulators. To obtain efficient phase modulation, it is essential to maximize the overlap between the optical mode and the RF electrical signal, quantified by the electro-optic overlap integral[36],

$$\Gamma = \frac{g}{V} \frac{\varepsilon_0 c n_{PZT} \iint_{PZT} E_x^e |E_x^{op}|^2 dxdy}{\iint Re(\mathbf{E}^{op} \times \mathbf{H}^{op*}) \cdot \hat{\mathbf{e}}_z dxdy}, \tag{1}$$

where $g$ is the spacing between the electrodes, $V$ the applied voltage, $\varepsilon_0$ the vacuum permittivity, $c$ the speed of light in vacuum, and $n_{PZT}$ the refractive index of PZT. $E_x^e$ is the in-plane ($x$-)component of the RF electric field, and $E_x^{op}$ represents the in-plane transversal component of the optical field. When used as a phase shifter, an important figure of merit is the half-wave voltage-length product $V_\pi L$. This product relates to the electro-optic coefficient $r_{eff}$ of the PZT films and to $\Gamma$[36],

$$V_\pi L = \frac{\lambda g}{n_{PZT}^3 \Gamma r_{eff}}, \tag{2}$$

where $\lambda$ is the free-space wavelength. Another important parameter is the propagation loss of the optical mode, consisting of an intrinsic contribution (scattering, material loss in the PZT, intermediate layer, nitride and oxide) and a contribution caused by the vicinity of the electrical contacts. The former can be estimated based on cut-back measurements on unmetalized waveguides (see Supplementary Note 2), and the latter can be numerically calculated.

**Data availability.** All data that support the findings of this study are available from the corresponding authors upon reasonable request.

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

## Acknowledgements

We thank Stéphane Clemmen for his overseeing role in the SiN chip fabrication, Philippe F. Smet for help with processing, Joris Van Kerrebrouck for help with data analysis, Liesbet Van Landschoot for operating the electron microscope, and Yoko Ohara for help with the figures. K.A. is funded by FWO Flanders. This work was funded by the European Commission through grant agreement no. 732894 (FET-Proactive HOT).

## Author contributions

K.A. and J.P.G. designed the devices. K.A. performed the chip planarization. J.P.G. performed the PZT deposition, patterning, and metalization. K.A. performed the static optical measurements. J.V. and K.A. carried out the high-speed measurements. K.A. analyzed the data and performed device optimization simulations. K.N., B.K., D.V.T.,

and J.B. provided general advice and feedback. K.A. and J.P.G. wrote the manuscript, and all authors reviewed the manuscript and agree to its content.

## Additional information

**Competing interests:** The authors declare no competing interests.

