## [Peer Review File · Nature Communications]

Reviewers' comments:

Reviewer #1 (Remarks to the Author):

The manuscript "Nanophotonic Pockels modulators on a silicon nitride platform" is very well written and reports novel and highly relevant results. I have no doubts that it meets the quality criteria for publication in Nature Communications.

I have a couple of minor comments that the authors may want to address. Point 17 should be mandatorily replied to, as there might be a typo in Eq. (2) making it inexact.

1. End of last paragraph: "...in other fields as well, such as nonlinear and quantum optics [5-7], microwave photonics [8], optical phased arrays for LIDAR or free-space communication [9] and more."

Please consider making communications plural, both because it expresses a category (which are usually plural, as in photonics) and to maintain consistency with the preceding categories, that are also all plural.

2. Paragraph 6, end of introduction. "O- and C-band". I believe it should be "O- and C-bands".
3. "These results not only strongly improve upon what is currently possible in SiN, they are also on par with the state-of-the-art modulators in silicon-on-insulator". The authors might want to differentiate this statement some. While the demonstrated $V_{\pi L}$ is still on the high side even compared to conventional PIN depletion modulators, the $V_{\pi L\alpha}$ is already excellent. Moreover "state-of-the-art modulators in silicon-on-insulator" could also be taken to include silicon-organic hybrid slot waveguide modulators, in which case the characteristics would not be that much on par... My recommendation would be to replace "state-of-the-art modulators in SOI" by "state-of-the-art carrier dispersion modulators" (which would be fair since SOH modulators are associated to other complications) and to differentiate between $V_{\pi L}$ and $V_{\pi L\alpha}$. If the authors believe it doesn't fit in the flow of their text at this point, I would recommend, at the very least, to highlight the excellent $V_{\pi L\alpha}$ down the line at the appropriate point.
4. Page 2, "Subsequently, PECVD SiO₂ (thickness ~ 1μm) is deposited over the devices and planarized, either by using a combination of dry and wet etching, or by chemical-mechanical polishing (CMP)". At this point it wasn't clear to me that the planarization was made such that the top surface would be flush with the upper edge of the SiN core (as opposed to the core to be still buried in a flat, but finite silica layer). I had to refer to the figures to visualize the structure. I recommend clarifying this in the text at this location.
5. "For the samples planarized through CMP, waveguide losses of around 1 dB/cm are measured". It wasn't clear to me at first whether this was with or without the PZT. If this is for the complete structure just without the metal electrodes, please clarify this in the text.

6. Losses at the optical transitions should be addressed somewhere in the manuscript. If the losses are still high because the authors haven't worked on optimizing transitions yet, that is fine from my perspective, but it should be stated and currently obtained losses quantified.
7. Page 3, "From this we estimate the half-wave..." Please consider putting a comma after "this".
8. "The poling stabilized and there are no indications of decay over much longer periods of time". I would recommend sticking to past tenses throughout, e.g., "The poling stabilized and there have been no indications of decay..."
9. The issue of long term stability is an important one indeed, particularly because the poling was made at room temperature. Any information that could be provided on this would be valuable, also in terms of previous long term stability studies reported in the literature if there are any...
10. Section "High speed characterization": "The bandwidths are not limited by the intrinsic material response of PZT." If this argument is based on known material properties from the literature, a reference would be in order here (also as an indication to the reader on what basis the argument is being made)...
11. Beginning of page 4: "are plotted as function of the PZT layer thickness, and the electrode spacing". Please consider replacing by "are plotted as a function of the PZT layer thickness and of the electrode spacing."
12. $V_{\pi L_{\alpha}}$ of 1 VdB is mentioned at the end of the paragraph without further comment. As mentioned above, I believe this to be excellent and that it should be highlighted. A major drawback of carrier dispersion modulators are their inherent losses that are absent from Pockels effect modulators – I believe this ought to be an important point.
13. Section "PZT deposition and patterning": "The details of the lanthanide-assisted deposition... have been published elsewhere, a short summary is given here" -> "While the details of the lanthanide-assisted deposition... have been published elsewhere, a short summary is given here"
14. Bottom of page 4: "for 15 min in tube furnace" -> "for 15 min in a tube furnace"
15. Eq. (1) on page 5: Note that this is the approximation of the overlap integral as applied to low index contrast systems (from scalar wave theory). Since the SiN / PZT platform is associated to substantial index contrasts, I would have applied the more rigorous formula associated to fully vectorial wave theory. I don't want to ask the authors to redo all their analysis on their basis, as it will probably not change the results of the analysis in a very meaningful way, I would however ask that they acknowledge the restricted validity of the formula to avoid it "propagating". Much of nonlinear optics textbooks have been written in the context of glass fibers and lithium niobate modulators (with doped waveguides), both low index contrast systems, so that it is

indeed common to use these less accurate formulas even in the silicon photonics literature.

16. "where λ is the wavelength": Please consider specifying as the free-space wavelength
17. Equation 2: I am a bit puzzled by n_{eff}^3 appearing in the formula instead of n_{PZT}^3 , with n_{PZT} the refractive index of the PZT material. The term n^3 comes from the definition of the *material* coefficient r , given by $\Delta(1/n^2)=r \cdot E=-2\Delta n/n^3$ resulting in $\Delta n = -n^3 r E/2$. Note that n is the refractive index of the nonlinear material, here n_{PZT} , and Δn is the refractive index change inside the non-linear material. The effective index change is then given by $-\Gamma V/g \cdot n^3 r E/2$, i.e., it is still n_{PZT} that appears in the equation. Maybe the swap between n_{PZT} and n_{eff} has something to do with a definition of r_{eff} ? That still doesn't make sense to me though, as all the other terms in Eq. (2) are well defined and unambiguous.

Reviewer #2 (Remarks to the Author):

The authors present a platform based on lead zirconate titanate thinfilms integrated with silicon nitride. They demonstrate high bandwidth modulators operating at 1310nm and 1550nm wavelengths.

The results are promising to add the missing ingredient to the silicon nitride photonic platform: electro-optic modulation.

The article is very well written. It is clear and well organized. I believe the manuscript should be accepted after addressing the following comments:

- The authors motivate the work by stressing the need for compatibility with CMOS fabrication techniques. However, the PZT material is annealed at 650C. Such a high temperature is only compatible with front end of the line processes and limits the applicability of the platform. Would this platform be compatible with the back end of the line processes? What is the performance if the highest process temperature is limited to 400C?
- While the experimental work looks to be at a very high level, I disagree in the inclusion of the 40gbps data. The eye diagram at 40gbps (Fig. 3c third panel) is essentially closed. The bit error rate would be very high at that speed. I understand that the modulation source is the limitation, but what does a closed eye add to the manuscript? In my opinion, the results at 40gbps should be omitted as they detract from the otherwise excellent results.

We have revised our manuscript thoroughly. First and foremost we would like to thank the reviewers for their detailed comments and fair review. We believe that the some important points were raised which enabled us to improve the manuscript. The detailed response to the points raised can be found below.

Reply to the reviewers:

Reviewer #1:

1. End of last paragraph: "...in other fields as well, such as nonlinear and quantum optics [5-7], microwave photonics [8], optical phased arrays for LIDAR or free-space communication [9] and more." Please consider making communications plural, both because it expresses a category (which are usually plural, as in photonics) and to maintain consistency with the preceding categories, that are also all plural.

The word "**communications**" has been made plural.

2. Paragraph 6, end of introduction. "O- and C-band". I believe it should be "O- and C-bands".

The suggested change has been made.

3. "These results not only strongly improve upon what is currently possible in SiN, they are also on par with the state-of-the-art modulators in silicon-on-insulator". The authors might want to differentiate this statement some. While the demonstrated $V_{\pi}L$ is still on the high side even compared to conventional PIN depletion modulators, the $V_{\pi}L\alpha$ is already excellent. Moreover "state-of-the-art modulators in silicon-on-insulator" could also be taken to include silicon-organic hybrid slot waveguide modulators, in which case the characteristics would not be that much on par... My recommendation would be to replace "state-of-the-art modulators in SOI" by "state of- the-art carrier dispersion modulators" (which would be fair since SOH modulators are associated to other complications) and to differentiate between $V_{\pi}L$ and $V_{\pi}L\alpha$. If the authors believe it doesn't fit in the flow of their text at this point, I would recommend, at the very least, to highlight the excellent $V_{\pi}L\alpha$ down the line at the appropriate point.

We agree that this part of the text needed some extra nuance. The point we want to make is is that when including the loss ($V_{\pi}L\alpha$), the modulators on this platform can be at least comparable, if not better than "established" silicon carrier dispersion modulators.

Indeed, due to the excellent $V_{\pi}L$'s achieved with organic hybrid slot waveguide modulators, the comparison there is less trivial, so we changed the terminology as suggested.

We also nuanced the comparison with other phase modulators on SiN somewhat, since the differentiating figure of merit there is the modulation speed (John Bowers' group recently demonstrated very good $V_{\pi}L\alpha$'s on SiN, but at much lower speeds - Jin, W. et al. *Piezoelectrically tuned silicon nitride ring resonator. Optics Express (2018)*).

The final part of the paragraph is rewritten as: **"These results, especially in terms of the achieved modulation bandwidths, strongly improve upon what is currently possible in SiN [25, 26]. In terms of $V_{\pi}L\alpha$, this platform can furthermore improve upon carrier dispersion modulators in SOI, which suffer from inherent carrier-induced losses absent in Pockels modulators [28]."**

To highlight the importance of the figure-of-merit $V_{\pi}L\alpha$, we also added (also see our reply to comment 12):

"Hence, the platform provides an excellent trade-off between optical losses and modulation efficiency. According to simulations, the product $V_{\pi}L\alpha$ can be as low as 2 VdB in optimized structures."

4. Page 2, "Subsequently, PECVD SiO₂ (thickness ~ 1 μ m) is deposited over the devices and planarized, either by using a combination of dry and wet etching, or by chemical-mechanical polishing (CMP)". At this point it wasn't clear to me that the planarization was made such that the top surface would be flush with the upper edge of the SiN core (as opposed to the core to be still buried in a flat, but finite silica layer). I had to refer to the figures to visualize the structure. I recommend clarifying this in the text at this location.

For clarification, we added the sentence **"This step is performed so that the SiN waveguide top surface and the surrounding oxide are coplanar."**

5. "For the samples planarized through CMP, waveguide losses of around 1 dB/cm are measured". It wasn't clear to me at first whether this was with or without the PZT. If this is for the complete structure just without the metal electrodes, please clarify this in the text.

This measurement was on PZT-covered spirals, without metallic contacts. For clarification, we changed the text to **"...propagation losses of around 1 dB cm⁻¹ are measured on PZT-covered waveguides without metallic contacts."** A similar sentence was added to Supplementary Note 2.

6. Losses at the optical transitions should be addressed somewhere in the manuscript. If the losses are still high because the authors haven't worked on optimizing transitions

yet, that is fine from my perspective, but it should be stated and currently obtained losses quantified.

We agree with this remark, indeed the coupling (grating plus coupling from bare SiN waveguide to PZT-covered section) has not been optimized so far. The following sentence was added:

“The combined loss of a grating coupler and the transition between a bare and PZT-covered waveguide section is \approx 12 dB at the optimum, with a 3 dB bandwidth of \approx 90 nm. However this is currently not optimized and can still be improved by design.”

7. Page 3, “From this we estimate the half-wave...” Please consider putting a comma after “this”.

The suggested change has been made.

8. “The poling stabilized and there are no indications of decay over much longer periods of time”. I would recommend sticking to past tenses throughout, e.g., “The poling stabilized and there have been no indications of decay...”

The suggested change has been made.

9. The issue of long term stability is an important one indeed, particularly because the poling was made at room temperature. Any information that could be provided on this would be valuable, also in terms of previous long term stability studies reported in the literature if there are any...

As far as we are aware, there are no further studies on the long term poling stability of thin film PZT.

10. Section “High speed characterization”: “The bandwidths are not limited by the intrinsic material response of PZT.” If this argument is based on known material properties from the literature, a reference would be in order here (also as an indication to the reader on what basis the argument is being made)...

This is an assumption based on theoretical grounds as presented in literature. For this we refer to “*Stefan Abel and Jean Fompeyrine (2016), Electro-Optically Active Oxides on Silicon for Photonics. Thin Films on Silicon: pp. 455-501.*” and in “*Peter Günter (1987) Electro-optical effects in dielectric crystals, Ferroelectrics, 75:1, 5-23*”. According to these references, the electro-optic effect in thin ferroelectric films has different contributions. In the frequency range of interest, two contributions are present, an electronic contribution and one governed by optical phonons, the slowest of these two, has a bandwidth on the order of 10^{12} Hz. Hence the electro-optic material response is not the limiting factor for our measured bandwidths (almost 2 orders of magnitude lower).

In the revised paper, these references have been added, and the following change has been made:

“The bandwidths are not limited by the intrinsic material response of PZT, but by device design and/or characterization equipment, **as the dominating contributions to the Pockels effect are expected to have a bandwidth which is almost two-orders of magnitude larger [29, 30].**”

11. Beginning of page 4: “are plotted as function of the PZT layer thickness, and the electrode spacing”. Please consider replacing by “are plotted as a function of the PZT layer thickness and of the electrode spacing.”

The suggested change has been made.

12. $V_{\pi}L\alpha$ of 1 VdB is mentioned at the end of the paragraph without further comment. As mentioned above, I believe this to be excellent and that it should be highlighted. A major drawback of carrier dispersion modulators are their inherent losses that are absent from Pockels effect modulators – I believe this ought to be an important point.

See response to remark 3. To highlight the excellent $V_{\pi}L\alpha$. The final paragraph of the introduction was adapted and the $V_{\pi}L\alpha$ value obtained in the optimization section was mentioned (the updated one, see our response to remark 17). We also highlighted the mentioned drawback of carrier dispersion modulators in the final sentence of this paragraph.

13. Section “PZT deposition and patterning”: “The details of the lanthanide-assisted deposition... have been published elsewhere, a short summary is given here” -> “While the details of the lanthanide-assisted deposition... have been published elsewhere, a short summary is given here”

The suggested change has been made.

14. Bottom of page 4: “for 15 min in tube furnace” -> “for 15 min in a tube furnace”

The suggested change has been made.

15. Eq. (1) on page 5: Note that this is the approximation of the overlap integral as applied to low index contrast systems (from scalar wave theory). Since the SiN / PZT platform is associated to substantial index contrasts, I would have applied the more rigorous formula associated to fully vectorial wave theory. I don’t want to ask the authors to redo all their analysis on their basis, as it will probably not change the results of the analysis in a very meaningful way, I would however ask that they acknowledge the restricted validity of the formula to avoid it “propagating”. Much of nonlinear optics textbooks have been written in the context of glass fibers and lithium niobite modulators (with

doped waveguides), both low index contrast systems, so that it is indeed common to use these less accurate formulas even in the silicon photonics literature.

See response to remark 17.

16. “where λ is the wavelength”: Please consider specifying as the free-space wavelength
The suggested change has been made.

17. Equation 2: I am a bit puzzled by n_{eff}^3 appearing in the formula instead of n_{PZT}^3 , with n_{PZT} the refractive index of the PZT material. The term n^3 comes from the definition of the material coefficient r , given by $\Delta(1/n^2)=r.E=-2\Delta n/n^3$ resulting in $\Delta n = -n^3 r E/2$. Note that n is the refractive index of the nonlinear material, here n_{PZT} , and Δn is the refractive index change inside the non-linear material. The effective index change is then given by $-\Gamma V/g \cdot n^3 r E/2$, i.e., it is still n_{PZT} that appears in the equation. Maybe the swap between n_{PZT} and n_{eff} has something to do with a definition of r_{eff} ? That still doesn't make sense to me though, as all the other terms in Eq. (2) are well defined and unambiguous.

This is a combined answer to remarks 15 and 17:

We did the derivation of the formulas from scratch, and we regret to say that there was indeed a mistake in the original manuscript!

We believe that Eq. (2) should indeed read:

$$V_{\pi} L = \frac{\lambda g}{n_{\text{PZT}}^3 \Gamma r_{\text{eff}}}$$

Our wrong formula was naively taken from literature and can be found in many papers:

Tang, Pingsheng, et al. "Low-voltage, polarization-insensitive, electro-optic modulator based on a polydomain barium titanate thin film." *Applied physics letters* 85.20 (2004): 4615-4617.

Himmelhuber, Roland, et al. "A silicon-polymer hybrid modulator—design, simulation and proof of principle." *Journal of Lightwave Technology* 31.24 (2013): 4067-4072.

Hu, Xuan, et al. "Modeling the anisotropic electro-optic interaction in hybrid silicon-ferroelectric optical modulator." *Optics express* 23.2 (2015): 1699-1714.

Xiong, Chi, et al. "Active silicon integrated nanophotonics: ferroelectric BaTiO₃ devices." *Nano letters* 14.3 (2014): 1419-1425.

We now believe that correct formulas can be found in for example the supplementary information of “*Koeber, Sebastian, et al. Femtojoule electro-optic modulation using a silicon–organic hybrid device. Light: Science & Applications (2015)*”, in accordance with this reference (but slightly different due to the inclusion of our g/V prefactor and the RF/DC field in the integral), we also get a more appropriate expression for the electro-optic overlap integral, Eq. (2) (assuming the material is primarily poled in the x-direction):

$$\Gamma = \frac{g \epsilon_0 c n_{PZT} \iint_{PZT} E_x^e |E_x^{op}| dx dy}{V 2P}$$

With $P = \frac{1}{2} \iint \Re(\mathbf{E}^{op} \times \mathbf{H}^{op*}) \cdot \hat{\mathbf{e}}_z dx dy$ the optical power in the mode.

In view of this we adapted both formulas and we have redone the calculations based on them. Due to the third-order dependence on the index, and a quite significant change in Γ , the changes to the derived values are significant. Also the analysis in sections “Device optimization” and the corresponding Supplementary Section “Device optimization - influence of the waveguide width” had to be redone.

Corresponding changes to the manuscript:

- **Methods section:** formulas and corresponding reference have been changed.
- **Extracted values of r_{eff} :** the values are now 61 pm/V, 67 pm/V and 70 pm/V, respectively for C-band ring, C-band MZI and O-band ring. The sentence “good comparison with ellipsometry measurements on our thin films” consequently had to be removed. We believe that this discrepancy is not that surprising, in the ellipsometry measurements, the PZT was poled out-of-plane, in the current measurement the material is poled in-plane at room temperature, hence it is more difficult to pole (consequently the poling efficiency seems to be something with extra margin of possible improvement).
- The section “**Device optimization**” and corresponding figure 4 have been changed, as the simulation presented in figure 4 had to be redone (using different equations, for a different waveguide width and for a different Pockels coefficient). The conclusions are that the devices can still be improved, albeit less drastic (minimum value of 2 VdB instead of 1 VdB). The supplementary section “**Device optimization - influence of the waveguide width**” had to be changed in the same way, resulting in a difference of the optimal waveguide width.
- **Change to the abstract:** in simulation, values of $V_{\pi}L$ below 1 Vcm are still achieved, however for waveguide parameters for which the optical loss becomes much too large, so it seems unfair to keep the sentence “Simulations indicate that values below 1Vcm are achievable.”, we replaced it by: “**Simulations indicate that further improvement is possible.**”

Reviewer #2:

1. The authors motivate the work by stressing the need for compatibility with CMOS fabrication techniques. However, the PZT material is annealed at 650C. Such a high temperature is only compatible with front end of the line processes and limits the applicability of the platform. Would this platform be compatible with the back end of the line processes? What is the performance if the highest process temperature is limited to 400C?

We agree with this comment, the high annealing temperature is one of the limiting factors of the technology. We believe that in our system no cristalization will occur below ~500C (anealing temperatures around 450C have been reported, however they rely on seed layers chosen specifically for this purpose, see for example "*Huang, Z. Et al. Low temperature crystallization of lead zirconate titanate thin films by a sol-gel method. Journal of Applied Physics (1999)*"). In our process, we would end up with amorphous PZT at annealing temperatures compatible with the back-end-of-line.

In the originally submitted paper, no explicit claim was made with of compatibility with back-end-of-line CMOS fabrication. However we agree with the importance of this point. To stress this we changed the last sentece of the 4th introctory paragraph:

"...enabling direct deposition of the layer on top of SiN waveguides fabricated using front-end-of-line CMOS processes."

2. While the experimental work looks to be at a very high level, I disagree in the inclusion of the 40gbps data. The eye diagram at 40gbps (Fig. 3c third panel) is essentially closed. The bit error rate would be very high at that speed. I understand that the modulation source is the limitation, but what does a closed eye add to the manuscript? In my opinion, the results at 40gbps should be omitted as they detract from the otherwise excellent results.

We agree that the 40Gbps shows a drop in quality. However the visual interpretation of eye diagrams is inherently subjective and very dependent on the way the eye is plotted (color scaling, overlapping points make it difficult to asses the point density, etc.). As the reviewer mentions the limit is the source and using a source that maintains a higher voltage swing at high bitrates (some papers do this, for example "*Wang, Cheng, et al. "Nanophotonic lithium niobate electro-optic modulators." Optics Express 26.2 (2018): 1547-1555.*") could create a different impression.

We would prefer to leave the 40Gbps eye diagram in the paper, and to leave the interpretation to the reader. To invigorate this argument, we estimated the bit error rate of our different eye diagrams, based on an estimation of the eye Q factor (as explained in “Agrawal, Govind P. *Fiber-optic communication systems*. John Wiley & Sons, 2012.”). We added the result of this analysis in the Supplementary Note 3.

Note that the estimated BER at 40 Gbps is about 3.2×10^{-3} , this is below the hard-decision forward error coding limit (HD-FEC) with 7% overhead of 3.8×10^{-3} , which is often used in literature, for example in:

Cho, Junho, Chongjin Xie, and Peter J. Winzer. "Analysis of soft-decision FEC on non-AWGN channels." Optics Express 20.7 (2012): 7915-7928.

Asif, Rameez. "Advanced and flexible multi-carrier receiver architecture for high-count multi-core fiber based space division multiplexed applications." Scientific reports 6 (2016): 27465.

Verbist, J., et al. "Real-time 100 Gb/s NRZ and EDB transmission with a GeSi electroabsorption modulator for short-reach optical interconnects." Journal of Lightwave Technology 36.1 (2018): 90-96.

Changes to the paper:

- Last paragraph of the “High-speed characterization” section: We added **“The bit error rates were estimated from the measured eye diagrams [31], and are below the hard-decision forward error coding (HD-FEC) limit of 3.8×10^{-3} [32,33] for bitrates up to 40 Gbps (see Supplementary Note 3).”**
- Extra supplementary section: **“Supplementary Note 3. Estimation of the bit error rate”**

Other changes:

A full data availability statement is included in the manuscript.

Captions for Figs. 1b and Supplementary Figs. 1c, 1f, 1i, were changed: The fact that nonlinear adjustments were used to enhance image contrast is now also mentioned.

Acknowledgements were added to Joris Van Kerrebrouck, Liesbet Van Landschoot and Yoko Ohara

The **model number of the oscilloscope** used for the eye diagrams was corrected in the section “Methods -> High-speed measurements”

Units have been changed according to the required formatting (e.g. pm V^{-1} instead of pm/V).

Supplementary information has been adapted to the required formatting.

Contributions statement was added.

Conflict of interest statement was added.

REVIEWERS' COMMENTS:

Reviewer #1 (Remarks to the Author):

Dear Authors,

thanks for addressing my comments. In particular, I now agree with Eqs. (1) and (2).

Dear Editor,

The authors have thoroughly addressed my comments. These are very nice results, in my opinion, and I recommend publication in Nature Communications.

Reviewer #2 (Remarks to the Author):

The revised manuscript is satisfactory. Thank you.